# RIO: A Benchmark for Reasoning Intention-Oriented Objects in Open Environments

**Mengxue Qu**[1,3*]  **Yu Wu** [4]  **Wu Liu** [5]  **Xiaodan Liang** [6,7]
**Jingkuan Song** [8]  **Yao Zhao**[1,2,3†]  **Yunchao Wei**[1,2,3†]

[1]Institute of Information Science, Beijing Jiaotong University   [2]Peng Cheng Laboratory
[3]Beijing Key Laboratory of Advanced Information Science and Network Technology
[4]Wuhan University  [5]JD Explore Academy  [6]Sun Yat-sen University  [7]MBZUAI
[8]University of Electronic Science and Technology of China
qumengxue@bjtu.edu.cn, wuyucs@whu.edu.cn, yunchao.wei@bjtu.edu.cn

## Abstract

Intention-oriented object detection aims to detect desired objects based on specific intentions or requirements. For instance, when we desire to "lie down and rest", we instinctively seek out a suitable option such as a "bed" or a "sofa" that can fulfill our needs. Previous work in this area is limited either by the number of intention descriptions or by the affordance vocabulary available for intention objects. These limitations make it challenging to handle intentions in open environments effectively. To facilitate this research, we construct a comprehensive dataset called Reasoning Intention-Oriented Objects (RIO). In particular, RIO is specifically designed to incorporate diverse real-world scenarios and a wide range of object categories. It offers the following key features: 1) intention descriptions in RIO are represented as natural sentences rather than a mere word or verb phrase, making them more practical and meaningful; 2) the intention descriptions are contextually relevant to the scene, enabling a broader range of potential functionalities associated with the objects; 3) the dataset comprises a total of 40,214 images and 130,585 intention-object pairs. With the proposed RIO, we evaluate the ability of some existing models to reason intention-oriented objects in open environments.

## 1  Introduction

Category-based object detection has been extensively investigated in recent years [1, 35, 5, 44], where common object categories are typically defined based on object names, such as "cat", "dog", "car", and so forth. These name-based categories facilitate the recognition and classification of objects with similar visual characteristics by computer systems. However, variations in category definitions among individuals pose challenges in predefining categories for all objects worldwide. In reality, individuals tend to search for objects based on their immediate needs rather than predefined category names. This challenges the traditional approach of category-based object detection, as the focus shifts towards objects that fulfill specific functional requirements.

Previous research [6, 32, 30, 47, 28] has explored the concept of "Affordance" to establish a connection between functional requirements and objects. "Affordance" refers to the function or potential action that an object offers for utilization. For instance, when seeking to rest, one may look for objects that afford the action of "lying down", such as a "bed", "sofa", or "bench". However, this approach can lead to ambiguity as the same verb may apply to multiple objects. For example, the verb "play" could refer to playing frisbee or playing the piano. Previous works use a limited number

---

*Work done during an internship at JD Explore Academy.
†Co-corresponding author.

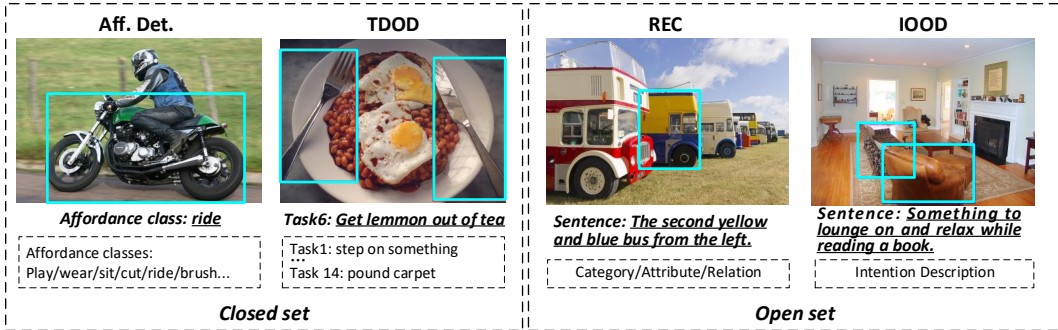

Figure 1: Comparison between tasks: Affordance Detection (Aff. Det.), Task-Driven Object Detection (TDOD), Referring Expression Comprehension (REC), and Intention-Oriented Object Detection (IOOD). **Closed set**: Aff. Det and TDOD are set to detect the object that matches the class or task of the limited closed set from the given image. There are limited affordance classes in Aff. Det. and 14 predefined tasks in TDOD. **Open set**: REC is set to detect the object that matches the referring expression mainly containing category, attribute, and relationship to other objects while there is no explicit category in IOOD. Only the intention description is given in IOOD.

of verb phrases to represent an intention, which fails to capture the diverse range of human intentions necessary to recognize the desired object accurately.

To address the issue of intention-oriented object reasoning in open environments, we introduce the RIO dataset in this paper, which is based on the images and annotated objects of the COCO dataset [20]. Compared to phrases, we consider sentences to be more descriptive and contain more relevant information. Our dataset contains multiple objects labeled with intention descriptions related to the scene. These descriptions are generated by conversing with ChatGPT and are comprehensive intention sentences rather than simple verb phrases, *e.g.*, the description for a frisbee is "Something you can use to play games with your dog.", while for a piano, the description is "Something you can use to create music.". The RIO dataset comprises 40,214 images and 130,585 intention-object pairs, where each intention-region pair refers to an intention description and its corresponding object in the image. It is worth noting that in our dataset similar intention descriptions may have different selected objects depending on the scene. For instance, in a park when the intention is to "lie down and rest", the objects like a "bench" or "grass" may fulfill the need rather than a "bed" or "sofa". This indicates that intention reasoning is challenging, which requires considering the environment and diverse intention descriptions rather than relying solely on category mapping and limited predefined intentions. In addition to this, we constructed an uncommon test set especially for some long-tail intentions, which are used for testing the model's few-shot capability.

We construct solutions for intention-oriented object reasoning in open environments based on existing methods and evaluate their capability with our proposed dataset. Our work contributes to the advancement of object recognition and reasoning by highlighting the significance of intention-based approaches in open environments. The RIO dataset provides a valuable resource for further research and development in intention-oriented computer vision tasks. Future work can explore the potential of incorporating context and environment reasoning into intention-based object recognition systems, leading to more robust and context-aware computer vision models.

## 2 Related Work

### 2.1 Affordance Detection

Affordance Detection is a critical component in intelligent systems as it empowers machines with the vital ability to understand and adapt to their environment and has important implications for human-robot interaction and machine decision-making. Affordance detection, on the one hand, is studied in physical machine systems and can facilitate robot-environment interactions. Grabner *et al.* [9] used 3D models of the human skeleton to learn how to sit on a chair and infer whether an object can provide the affordance of "sitting". On the other hand, affordance detection has also been studied in the field of vision only. Early work perceived affordances by establishing associations between object surface features and affordances, and Myers *et al.* [32] proposed a framework to

Table 1: Comparison with previous affordance detection datasets and task-driven object detection datasets. Our descriptions are natural sentences while others are words or phrases, and our descriptions are scene-related. The number of our affordance vocabulary is much higher than other datasets.

| Dataset | ADE-Aff | PAD | PADv2 | PAD-L | COCO-Tasks | Ours |
|---|---|---|---|---|---|---|
| # images | 10,000 | 4,002 | 30,000 | 4,002 | 39,724 | **40,214** |
| # categories | **150** | 72 | 103 | 72 | 49 | 69 |
| intentions | verb words | verb words | verb words | phrases | phrases | **sentences** |
| # affordance | 7 | 31 | 39 | 31 | 14 | > 100 |
| scene-related | × | × | × | × | × | ✓ |

jointly localize and identify the affordances of objects and presented the first pixel-level labeled affordance dataset.

However, the affordance of an object does not only correspond to surface features, it is also closely related to human interaction. Therefore, to address the practical problem of affordance inference, Chuang *et al.* [6] considered the physical world and social norms and constructed the ADE-Affordance dataset [6] based on ADE20k [49]. In reality, affordance is related to the purpose of human behavior, and considering the existence of multiple potential complementarities between animals and their environment as defined by Gibson, which leads to multiple possibilities for specific affordances, Luo *et al.* [30] constructed a purpose-driven affordance dataset PAD [30] and PADv2 [47] that considers the relationship between human purpose and affordance, covers more complex scenarios, extends the number of affordance categories, and is more applicable to practical robotic applications.

The aforementioned work focuses on computer vision. Further, Lu *et al.* attempted to investigate affordance detection involving natural language, *i.e.*, inputting a set of phrases and images describing affordances with the expectation of being able to segment out the corresponding objects. This is consistent with the reality that robots receive information from multiple sources. Thus the PAD-L dataset [28] is constructed based on the PAD dataset, in which the phrase collection is based on a limited affordance dictionary paraphrase. However, this form of expression is still not natural enough to match the natural human daily language description habits. In addition, people not only consider affordances when selecting objects but for objects with similar affordances, depending on the specific task, people will select targets that better match the task. Therefore, Sawatzky *et al.* [37] proposed task-oriented object detection and constructed the COCO-Task dataset based on the COCO dataset. In the COCO-Task dataset, people select the best class of objects in the images for different tasks. However, this dataset only predefined 14 tasks and used phrase representation, which lacked diversity.

Therefore, in this paper, we tried to reason about the corresponding objects using a more natural form of expression than affordance phrases, *i.e.*, sentences of intention description. We argue that predefined categories of affordances limit more possibilities for human-object interaction, while open-world natural sentences can cover more affordance situations.

## 2.2 Referring Expression Comprehension

Benefiting from the development of large-scale visual language pre-training models, many transformer-based models have been proposed, such as VL-BERT [38], VilBERT [27], UNITER[4], 12in1 [26], and MDETR [14]. In addition to REC, there has been a lot of research [26] related to the Referring Expression Segmentation (RES) task, which is to generate a binary mask based on the referring expression. Recently, a simple and universal network term SeqTR [50] is presented, which regards both REC and RES as a point prediction problem. SeqTR greatly reduces the difficulty and complexity of both architecture design and optimization. Based on SeqTR, Polyformer extends the sequence prediction of the mask when there are multiple contour polygons. Furthermore, by randomly downsampling dense contour points during training, the model can predict better contours with adaptive fitting in evaluation.

Previous work has more fully investigated the noun-based REC model [10, 11, 21, 39, 40, 41, 45, 48, 51, 3, 19, 36, 43, 42] and the RES model [29, 7, 8, 2, 12, 13, 17, 50, 7, 34], and great progress has been achieved in terms of performance. However, in real-world applications, such as intelligent service robots, system input is typically in the form of affordances (*i.e.*, the ability to support an action or speak a verb phrase). Whether modern visual language models are designed to effectively understand verb references remains under-explored. Li *et al.* [18] proposed TOIST to distill the noun

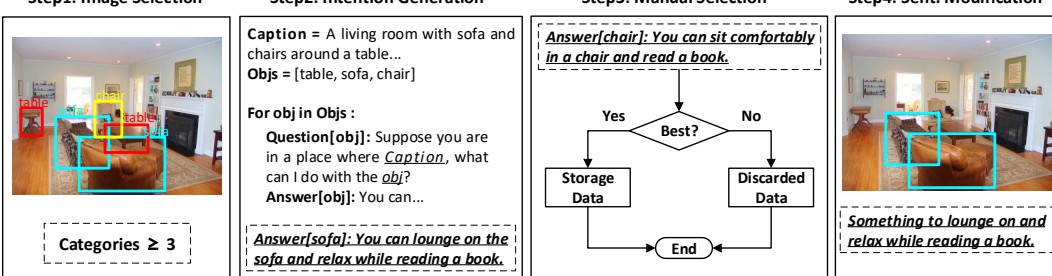

Figure 2: Step-by-step diagram of data collection. We first select those images with more than 3 categories of instances, then we chat with GPT to generate the draft of the intention description. For the draft description, we manually determine whether it is the best match and finally perform sentence conversion.

knowledge learned from REC models into verb comprehension models to solve Task-oriented Object detection and segmentation tasks.

In our work, with diverse descriptions of intentions, we validate the ability of TOIST models to reason about intention-oriented objects in the open environment. Also, we build baselines based on several recent REC and RES models to test their capabilities on our proposed RIO dataset.

## 3 RIO Dataset

In this section, we present the process of collecting the RIO dataset in Sec. 3.1, and then we provide statistics on the dataset distribution in Sec. 3.2. In Sec. 3.3, we propose evaluation metrics for the Intention-Oriented Object Reasoning task.

### 3.1 Dataset Collection

Our annotations are based on images and instance annotations from the COCO2014 dataset. The process of data collection is illustrated in Fig. 2. We collected two types of intention description annotations, called common and uncommon.

#### 3.1.1 Collection of Common Intentions

**Step1: Image Selection** We select the categories in the COCO dataset that are suitable for Intention-Oriented Object Reasoning, *i.e.*, categories with more affordances. We remove the category "People" because "People" is the one that generates Intention. At the same time, since the affordance of "food" is usually only "eat" and as an ingredient, we also remove the class related to "food". We want to interact in a complex open environment to collect more diverse descriptions of human intent, so we filter the image scenes from the COCO image set to include instances of more than 3 classes.

**Step2: Intention Generation** We generate intention descriptions for each category of instances in each image. We do not consider which object in the same category is more consistent with the intention description, but only which category of objects in the image is more consistent with the intention description. For example, as shown in Fig. 2, although there are multiple tables and sofas in the image, we only distinguish between categories when generating intention descriptions in *Step2*.

Due to the development of LLMs, ChatGPT is able to simulate human intentions and interactions to a certain extent, so we collect a draft description of intentions by talking to ChatGPT. In order to make ChatGPT "see" the image content, we first provide a **Caption** describing the content of the image, and then we ask ChatGPT *"what interaction it can have with a certain category of objects"*. Using the gpt3.5 model, we ask ChatGPT questions in the following format:

> *Question: "Suppose you are in a place: [Caption], What can you do with the [Obj]?*

**Step3: Manual Selection** We present the annotated images, instances, and intention description drafts from the previous steps to human annotators, who are asked to check the clarity of the intention statements and remove those intention descriptions that are irrational and unconventional. In addition, they are also asked to determine whether the intention and the object are the best match and to remove

*(a) Something you can use to cut the cake into slices.*  *(b) Something you can use to scoop the grilled fixings onto the buns.*  *(c) Something you can use to cut the strawberries and chocolate into small pieces.*  *(d) Something you can use to cut the pizza into smaller pieces.*

Knife -- Cut    Spoon -- Scoop    Spoon -- Cut    Fork -- Cut

Figure 3: Uncommon Intentions. We additionally collected some uncommon intents. for example, "cut" is an uncommon intention for spoons and forks.

those data pairs that are not the best match. Since we ask category-by-category during the collection process, there may be two categories of objects with similar intention descriptions, although the sentences may not be identical. *E.g.*, for the intention description "You can sit comfortably in a chair and read a book", there is a "sofa" in the diagram to sit more comfortably, so the original intention description of the chair should be deleted. Since human priority judgments are also more subjective, we only do careful filtering for the val and test sets. For the train set, the per-class intention description is retained, but we observe that there are also certain statistical properties in the train set, for which the more matching intention-instance pair appears more often.

**Step4: Sentence Modification** In practice, we should not use "You can use the **[obj]** to ..." to express our intention directly, but "I need **something** to ..." or "**Something** I can use to ...". Therefore, we automatically modify the sentence to remove the category name contained in the sentence.

### 3.1.2 Collection of Uncommon Intentions

We find that for some categories, there may be some less-used affordances. They may be used as a substitute for some other objects to perform some functions commonly used by other objects. As shown in Fig. 3, knife usually corresponds to cut, and spoon often corresponds to scoop. Spoons and forks are not generally used to cut things, but they do have the ability to cut soft things.

First, we analyze the intention descriptions of various categories to identify potential uncommon category-intention pairs. Subsequently, we create category mapping charts that suggest possible replacements for each other. We asked ChatGPT questions as follows:

> *Question: "Suppose you are in a place where: **[caption]**. Can the **[original class]** be used as a **[substitute class]**? Please answer yes or no. If yes, what can you do with the **[original class]**?*

For the spoon, the question is like *"Suppose you are in a place where: **[caption]**. Can the **spoon** be used as a **knife**? Please answer yes or no. If yes, what can you do with the **spoon**?"*, and the answer generated by ChatGPT is "You can use the spoon as a knife to cut the strawberries and chocolate into small pieces.". The final intention description is "Something you can use to cut the strawberries and chocolate into small pieces.".

Although we pre-define the category maps for mutual substitution, by asking ChatGPT in this format, we still get open-intention descriptions with diverse language expressions.

### 3.2 Dataset Statistics

Our dataset consists of 40,214 images and 130,585 object-intention description pairs. Our training set images are from the COCO train and account for 33% of the original COCO [20] train set, and the test set images are from COCO val and account for 30% percent of the original dataset. There is no overlap between COCO training images and our test images, so models pre-trained on COCO can be used and evaluated fairly. The test set is divided into two parts according to common and uncommon, and the images they use are overlapping, differing in the annotation of intentions. The final dataset contains 96,415 data pairs (27,696 images) for training, 29,344 data pairs (12,392 images)

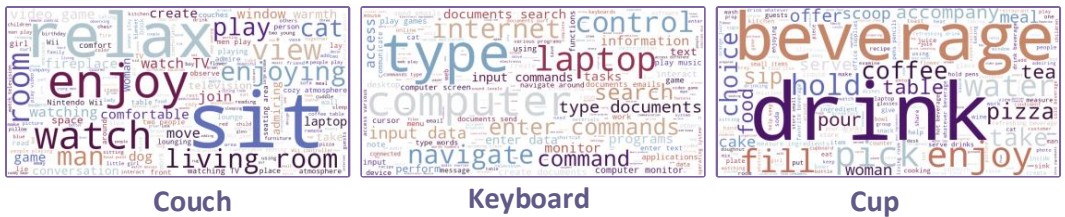



**Couch**  **Keyboard**  **Cup**



Figure 4: Word clouds of partial categories from the common set.

for common tests, and 4,248 data pairs (4,826 images) for uncommon tests. Common intentions are more consistent with the distribution of the train set, and uncommon intentions are an expansion of the common long-tail data, to test whether the model can few-shot after training with various intentions. Fig. 4 shows some word clouds and Fig. 5 illustrates the statistics of the relationship between categories and verbs. For each category, an average of 158 verb words appear within their descriptive sentences. There are about 300 (20%) verbs that appear in the intention descriptions of more than 10 categories, and thus cannot be detected by constructing a verb-noun mapping table, but must instead reason the target by intention, which makes the ROI task more challenging.

### 3.3 Evaluation Metrics

The task of intention-oriented reasoning is to locate objects that satisfy the intent given an intention description sentence. In order to facilitate the study of downstream application scenarios, we divide the task into two subtasks, intention-oriented object detection (IOOD) and intention-oriented instance segmentation (IOIS). We define different metrics for IOOD and IOIS in order to comprehensively evaluate the model performance.

**Intention-oriented object detection** We use Top1-Accuracy and AP@0.5 as evaluation metrics. Top1-Accuracy indicates that the model only needs to predict one correctly to understand the intention. We calculate the IoU (Intersection over Union) between each prediction bounding box and the ground truth bounding box and set the IoU $threshold > 0.5$ for correct prediction. However, we also want the model to have the ability to recall all objects that satisfy the intention, so we use the AP@0.5 object detection evaluation metric of COCO detection.

**Intention-oriented instance segmentation** We use Top1-IoU and mIoU as evaluation metrics. Similar to IOOD, Top1-IoU only calculates the IoU between the highest probability predicted object and the ground truth object masks. While mIoU (mean Intersection over Union) is the average of the IoU calculated for all preferred categories of objects.

## 4 Experiments

### 4.1 Construction of Detector-Reasoner Baseline

A straightforward baseline approach is to conceptualize the Intention-Oriented Object Detection (IOOD) task as a combination of Object Detection and Question Answering (QA). This can be realized by architecting a hybrid model comprising an object detector paired with an LLM (Language Language Model) reasoner. In the training of IOOD, the goal is to predict the object in an open environment based solely on the open-set intention description, without predicting predefined categories. To ensure the fairness of the comparison, we use the open-set object detection SOTA model, GroundingDINO [23], which is also class-agnostic, as the object detector. We directly utilize ChatGPT as our Language Reasoner.

GroundingDINO is trained without being confined to a specific category range, enabling it to detect objects from any language input. For the IOOD task, we utilize the COCO category list as input, facilitating zero-shot inference in GroundingDINO. Then we set a score threshold to filter the box predictions. This results in a list of candidate object bounding boxes represented as $[box_1, box_2, ..., box_n]$, and their associated category labels, denoted $[label_1, label_2, ..., label_n]$. With this information, we craft a prompt for ChatGPT structured as *"In categories $[label_1, label_2, ..., label_n]$, which category can you use to...?"*. ChatGPT subsequently selects the appropriate category. The final output from the hybrid system is the bounding box of the selected category.

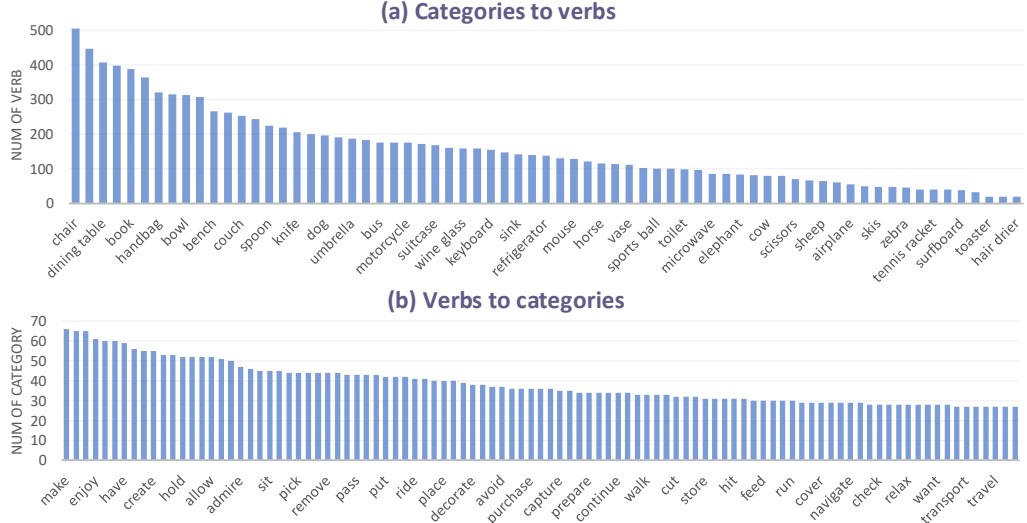

Figure 5: Dataset Statistics. (a) shows the statistics of the verb diversity in intention descriptions for each category, and (b) shows the frequency of occurrence of a verb in different categories of intention description. The horizontal coordinates are only examples of some categories and verbs.

## 4.2 Construction of End-to-End Baselines

In REC and RES tasks, there are many models and paradigms that primarily focus on noun-oriented reasoning and lack exploration of the understanding of verb-oriented intention. Since our task is similar to REC and RES, we construct baselines based on several state-of-the-art methods, *i.e.*, MDETR [14], SeqTR [50] and Polyformer [22] in REC and RES to evaluate their capability in verb-oriented intention reasoning. Additionally, a recent work TOIST [18] investigates task-oriented object detection and segmentation, and we also build a baseline based on it. In this section, we briefly review four related models and provide a detailed description of the baselines built upon them.

**MDETR** [14] is a Vision-Language (V-L) pre-training model which can achieve good performance after finetuning on VQA, REC, and other downstream V-L tasks. In pre-training, the training goal is set as follows: input a sentence and an image, make the model detect the objects corresponding to the nouns involved in the sentence, and draw the bounding boxes in the image. The object list corresponds one-to-one with the noun list. While in the task of intention-oriented reasoning, the form of the sentence is like *"something you can use to ..."*, we construct a correspondence between the objects and the word "something" only. The original model and the loss function remain unchanged, in this way the model can be trained to recognize objects referred to by the intention description.

**TOIST** [18] is an improved model based on MDETR. It is designed to solve task-oriented object detection and segmentation tasks. Considering that the trained MDETR has the noun recognition ability that may be useful for verb recognition, an attempt is made to retain this ability by noun-pronoun distillation in TOIST. Therefore, following TOIST, we first train the model with sentences with category nouns, *e.g.*, *"You can use the **knife** to cut cakes."*, then distillate to reason *"You can use something to cut cakes."*.

**SeqTR** [50] is the first work to unify both REC and RES as sequence prediction. In previous works, both MDETR and TOIST train REC and RES by detection head and segmentation head respectively, but in SeqTR, REC is defined as predicting the coordinates (x, y) of the top-left and bottom-right points of the bounding box, and RES is defined as predicting the sequence of contour points of the binary-mask. It is well worthwhile to test how this latest sequence prediction paradigm performs on IOOD tasks. Since this model can only predict a single object, in order to be able to roughly evaluate the ability of the sequence prediction model, we change the training target to predict the largest object that can satisfy the intention, and we only measure Top1-Accuracy and Top1-IoU when testing.

**Polyformer** [22] improves on the base of SeqTR. Considering that many objects are divided into multiple parts due to occlusion or their own shapes, Polyformer, a model that can be used for multiple polygon contour point prediction, is constructed. At the same time, this ability to predict multiple

Table 2: Comparison of baselines. "D-R" denotes the Detector-Reasoner hybrid model. Both D-R and D-R* employ GroundingDINO [23] as their object detector. D-R uses parameters from the GroundingDINO model that have not been fine-tuned on the COCO dataset, while D-R* uses parameters that have been fine-tuned on the COCO dataset.

| Method | Object Detection | | | | Instance Segmentation | | | |
| --- | --- | --- | --- | --- | --- | --- | --- | --- |
| | common | | uncommon | | common | | uncommon | |
| | AP@0.5 | Top1-Acc | AP@0.5 | Top1-Acc | mIoU | Top1-IoU | mIoU | Top1-IoU |
| D-R | 38.30 | 58.21 | 10.67 | 33.07 | - | - | - | - |
| D-R* | 45.15 | 64.77 | 17.25 | **41.98** | - | - | - | - |
| MDETR [14] | 48.61 | 65.05 | **24.20** | 39.60 | 44.14 | 54.55 | 22.03 | **34.35** |
| TOIST [18] | **49.05** | 66.72 | 21.96 | 39.28 | 45.07 | **55.85** | 19.41 | 34.00 |
| SeqTR [50] | – | **67.44** | – | 23.24 | – | 47.26 | – | 29.91 |
| Polyformer [22] | – | – | – | – | **48.75** | – | **26.77** | – |

polygons can be used exactly for inference when the intention description corresponds to multiple instances. We train and test Polyformer to reason intention-oriented objects on the RIO dataset while keeping the model and loss function unchanged.

## 4.3 Implementation Details

To make the most of the pre-trained noun recognition, we train all four models based on pre-trained parameters, which are trained on the merged region descriptions from Visual Genome (VG) [16] dataset, annotations from RefCOCO [46], RefCOCO+ [46], RefCOCOg [31], ReferItGame [15], and Flickr entities [33]. The total pretraining data is approximately 6.1M distinct language expressions and 174k images in the train set. The learning rate and backbone used for each model are basically the same as the original model, with some slight adjustments that are detailed in the supplementary material. All of our experiments are completed on four RTX 3090s.

## 4.4 Comparison of Baselines

As shown in Table 2, we report the results of four models on intention-oriented object detection and instance segmentation, including common set and uncommon set. Since SeqTR is a prediction model for a single object, we calculate Top1-Accuracy and Top1-IoU according to the only prediction result which is regarded as the highest probability. As for the multiple polygon prediction model Polyformer, we only report the mIoU results because there is no probabilistic prediction for multiple polygons and the output order of each object is random.

In terms of the results, end-to-end methods are superior to the Detector-Reasoner hybrid approach. This superiority can be attributed to inherent performance limitations associated with object detectors. For end-to-end methods, we can observe that the sequence prediction model SeqTR has a higher Top1-Acc than both MDETR and TOIST in the IOOD, but it can only recall one object. The noun-pronoun distillation introduced by TOIST makes some performance improvements compared to MDETR in the common set, but the performance degrades in the uncommon set. In the intention-oriented instance segmentation, the multi-polygon sequence prediction model Polyformer has a higher mIoU than MDETR and TOIST, and SeqTR has a lower Top1-IoU than MDETR and TOIST.

The same phenomenon in both tasks is that the performance is acceptable on the common set, but the performance drops very much on the uncommon set. This indicates that the model's ability to few-shot is relatively weak, and future work should mainly explore how to use common data training to improve the inference ability on uncommon data.

## 4.5 Ablation Study

In this section, we first analyze the advantages of end-to-end methods over hybrid methods. In addition, we perform ablation studies on MDETR and TOIST model structures and investigate the effect of sentence forms and distillation on the model.

Table 4: Ablation studies in MDETR [14] and TOIST [18] model structure. "Sent. Form" is Sentence Form.

| Model | Sent. Form | Pretrain | Confidence | AP@0.5 |
|---|---|---|---|---|
| (a) MDETR | "You can use something to ..." | ✗ | 'something' | 2.57 |
| (b) MDETR | "You can use something to ..." | ✓ | sentence | 27.6 |
| (c) MDETR | "You can use something to ..." | ✓ | 'something' | 48.16 |
| (d) MDETR | "You can use [obj] to ..." | ✓ | '[obj]' | 32.64 |
| (e) TOIST | "You can use [obj]–>something to ..." | ✓ | 'something' | 49.00 |

### 4.5.1 Advantage Analysis of End-to-End Approach over Hybrid Approach

**Performance Advantages** In terms of results, visual-text alignment is superior to the Detector-Reasoner hybrid approach. The final performance of the hybrid method can be affected by both bounding box accuracy and category accuracy. Misclassification of categories can lead to incorrect judgments when integrating with LLM.

**Generalization Advantages** One perennial challenge in open-set object detection is the inconsistent labeling of the same object by different annotators. This inconsistency arises because various annotators might use different category names for the same object, making standardization a complex task. To illustrate that, we conducted a simple ablation study. By replacing original COCO category terms with commonly used synonyms, the performance of GroundingDINO on the

| Category Name | AP | AP50 |
|---|---|---|
| Original | 48.4 | 64.6 |
| Synonymous | 35.6 | 47.5 |

Table 3: Simple ablation of categories with synonym replacement.

COCO2014 validation set exhibited a noticeable decline, as shown in Table 3. This observation emphasizes the motivation behind our IOOD task - our goal is to avoid intricate name definitions. Instead, the IOOD task focuses on detecting objects based on their functions and human intent, which encourages the model to understand the essence of the object, not just its categorical label.

**Structural Advantages** Compared with end-to-end models, hybrid models are limited by the capabilities of their individual components and often struggle to utilize the combined feature information from vision and language. Furthermore, in hybrid models, there's an increased risk of error propagation. In contrast, end-to-end systems streamline these processes, optimizing the synergy between vision and language while reducing cascading errors.

### 4.5.2 The Effect of Model Structure and Sentence Forms

As indicated in Table 4, comparing (a) and (c), the performance is significantly degraded when the model is not pre-trained. This shows that the ability of noun object detection learned during pre-training is helpful.

As we know in MDETR, the target of pre-training is to predict the object corresponding to each noun contained in the sentence. When MDETR is finetuned on the REC task, it shares the confidence weight equally for each word in the sentence because the whole sentence refers to an object. However, this approach may not be suitable in RIO because our sentences are much longer compared to the sentences in REC. Therefore, we set the confidence weight to be applied only to the word "something" instead of sharing it equally throughout the sentence. Comparing (b) with (c), we found that using the confidence of only the word "something" instead of the whole sentence, the performance improves. In (d), we introduce the object noun **[obj]** into the sentence, and the detection accuracy is indeed enhanced compared to the equivalent condition (b). (e) indicates the performance gain of using noun-pronoun distillation, *i.e.* distillation from "[obj]" to "something".

### 4.6 Qualitative Result

We visualize some of Polyformer's prediction results as examples in Fig. 6. (a)-(c) are the results of the common set, and (d) is the results of the uncommon set. The bounding box in the figure is of all instances in the Polyformer model, not the per-instance bounding boxes in the intention-oriented object detection task. The model can generally predict objects that match the intention description, and can also identify the uncommon intention.

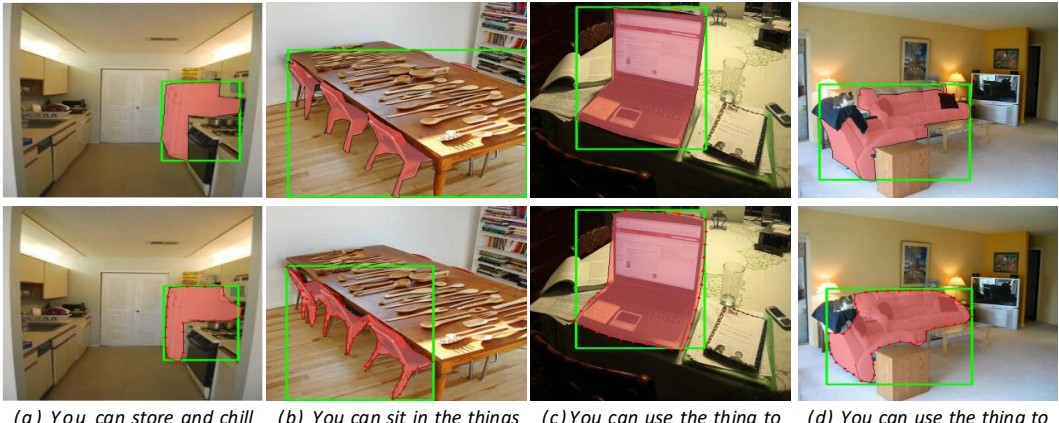

*(a) You can store and chill food and drinks in the thing.*  *(b) You can sit in the things to cook and eat at the table.*  *(c) You can use the thing to view documents, browse the internet, and use various programs and software.*  *(d) You can use the thing to fold out the bottom cushion and lay down a mattress or other sleeping surface.*

Figure 6: Visualization of Polyformer in IOIS. (a)-(c) are the results of the common set, and (d) is the results of the uncommon set. The first line is the ground truth, and the second line is the prediction polygons. The bounding box in the figure is of all instances in the Polyformer model, not the per-instance bounding boxes in the intention-oriented object detection task.

## 5    Conclusion and Discussion

In conclusion, this paper addresses the limitations of category-based object detection and introduces the RIO dataset, which focuses on intention-oriented object reasoning in open environments. The dataset consists of images from the COCO dataset and includes multiple objects labeled with comprehensive intention descriptions. Unlike simple verb phrases, these descriptions capture diverse human intentions related to the scene. The dataset contains a large number of intention-object pairs, allowing for robust evaluation of intention reasoning algorithms. By leveraging existing methods, we constructed solutions for intention-oriented object reasoning and evaluated their performance using the RIO dataset. The results demonstrate the importance of considering diverse intention descriptions and reasoning about the environment when recognizing the desired object accurately. This research contributes to the advancement of object recognition and reasoning by highlighting the significance of intention-based approaches in open environments. The RIO dataset provides a valuable resource for further research and development in intention-oriented computer vision tasks. Future work can explore the potential of incorporating context and environment reasoning into intention-based object recognition systems, leading to more robust and context-aware computer vision models.

## Acknowledgments and Disclosure of Funding

This work was supported by the Fundamental Research Funds for the Central Universities (2023JBZD003), the National NSF of China (No.U1936212, No.62120106009), the Fundamental Research Funds for the Central Universities (No. K22RC00010), and Beijing Nova Program (NO. 20220484063).

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

# Appendix

## A  Discussion

### A.1  General Discussion

**Limitation** Since we use ChatGPT to generate intention descriptions for each class of instances, the diversity and reasonableness of the intention descriptions depend on the performance of ChatGPT, and the manual filtering stage only filters inappropriate intention descriptions without the modification of intention descriptions. Meanwhile, the subjectivity of object usage for tasks can lead to ambiguity in determining task fulfillment criteria. Although we have provided clear guidelines to annotators and conducted multiple quality checks to ensure consistency, there still remains some subjectivity inconsistency.

**Ethical Concerns** Our dataset is constructed based on the existing publicly available dataset MSCOCO [20], so there are fewer privacy concerns. For the object instances in the images that are already labeled, our RIO dataset only supplements the available intention description of the instances. Our process of collecting the available intention of instances is a combination of automated and manual, where the construction of most of the matching sentence is automatic and only requires manual prioritization based on human preferences. Overall our approach is relatively environmentally friendly and less costly. Since the intention description is generated from conversations with ChatGPT, it may be affected by the bias of the ChatGPT training data, though it can be filtered out in the manual confirmation stage.

**License** MDETR [14], SeqTR [50], and PolyFormer [22] follow Apache License 2.0, TOIST [18] follow MIT License follow Apache License 2.0. We used author-released code for baseline construction. MSCOCO [20] annotation data is licensed under a Creative Commons Attribution 4.0 International License. In addition, ChatGPT is an AI language model developed by OpenAI, which is based on GPT-3.5 architecture. Specific licensing information for the GPT-3.5 model is a trade secret of OpenAI and therefore precise licensing details cannot be provided.

### A.2  Comprehensive Discussion about the Use of ChatGPT

**Quality** ChatGPT has been trained on a vast amount of text data, enabling it to generate high-quality text outputs. We randomly sampled 500 original answers generated by ChatGPT and assessed their suitability for dataset construction based on accuracy, completeness, and fluency. The usability rate of the data was 96.2%.

**Diversity** When using ChatGPT, the value of temperature can be adjusted to control the randomness of the output. The higher the value (closer to 1), the more varied and creative the output; the lower the value (closer to 0), the more deterministic but also potentially too repetitive the output. We set temperature=0.7 to ensure the diversity of the model output. In addition, WordCloud in Figure 4 and the dataset statistics in Figure 5 also illustrate the diversity of responses generated by ChatGPT.

**Human-likeness** ChatGPT is already been implemented as a product and, from a language expression perspective, it can provide relatively accurate descriptions of human intentions. From the perspective of linguistic naturalness, the 500 answers randomly sampled from original answers generated by ChatGPT all meet the standards of human likeness.

## B  Implementation Details

To make the most of the pre-trained noun recognition, we train all four models based on pre-trained parameters, which are trained on the merged region descriptions from Visual Genome (VG) [16] dataset, annotations from RefCOCO [46], RefCOCO+ [46], RefCOCOg [31], ReferItGame [15], and Flickr entities [33]. The total pretraining data is approximately 6.1M distinct language expressions and 174k images in the train set. The learning rate and backbone used for each model are basically the same as the original model, with some slight adjustments that will be detailed in the supplementary material. All of our experiments were completed within four RTX 3090s but with different learning rates and training epochs. The details are as follows.

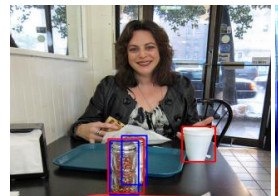 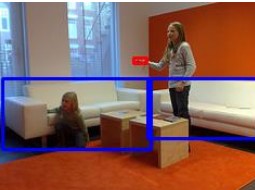 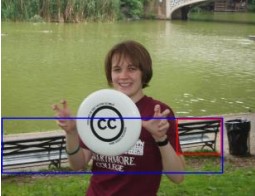 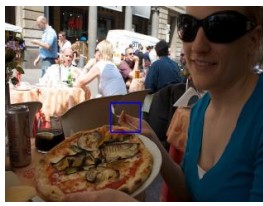

*(a)Something you can use to fill with water and hold flowers or other decorates.*  *(b)Something you can use to flip it into a sofa.*  *(c)Something you can use to provide a stable surface for writing or eating.*  *(d)Something you can use to cut the pizza into slices.*

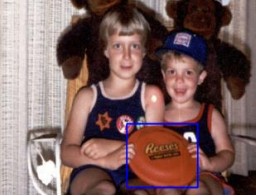 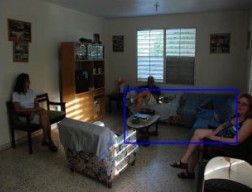 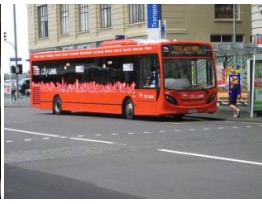 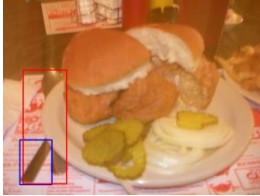

*(e)Something you can use to serve snacks and treats to the boys and animal friends.*  *(f)Something you can use to turn it into a makeshift bed by adding pilows.*  *(g)Something you can use to hold it above your head to keep you dry when raining.*  *(h) Something you can use to cut the sandwich and other items on the plate.*

Figure 7: The failure cases predicted by the TOIST model on the RIO dataset uncommon set, the blue box is the ground truth, and the red box is the prediction result. When no object in the model prediction result reaches the probability threshold, the output is the empty set.

**MDETR** Following MDETR [14], all parameters in the network are optimized using AdamW [25] with the learning rate warm-up strategy. The model is trained with a batch size of 80. We set the learning rate of the language backbone RoBERTa [24] to be $1 \times 10^{-5}$, and all the rest parameters to be $5 \times 10^{-5}$. In initial training, the model is trained for 30 epochs. We randomly resize and crop the image, and set the maximum side length of the input image as 640 while keeping the original aspect ratio. Images in the same batch are padded with zeros until acquiring the largest size of that batch. Similarly, sentences in one batch will be adjusted to the same length as well.

**TOIST** Following TOIST [18], both the student and teacher TOIST models are initialized with the model pre-trained by MDETR [14] and use the AdamW [25] optimizer. The batch size is set as 80. We fine-tune the teacher model on our RIO dataset for 30 epochs. Then we use the fine-tuned teacher model to distill knowledge to the student model for 15 epochs. The initial learning rates are set to $1 \times 10^{-5}$, $1 \times 10^{-5}$, $5 \times 10^{-5}$ for the language encoder, visual encoder, and other parts of the model.

**SeqTR** Following SeqTR [50], all parameters in the network are optimized with AdamW [25], and the batch size is 128. We load the pre-trained parameters and finetune the model with 30 epochs. The learning rate is set as $5 \times 10^{-4}$. Images are resized to $640 \times 640$, and the length of language expression is trimmed at 20. We set the number of sampled contour points as 18 points.

**PolyFormer** Following PolyFormer [22], we use the AdamW optimizer [25]. The initial learning rate is $5 \times 10^{-5}$ with polynomial learning rate decay. We load the pre-trained parameters on combined large-scale visual grounding data and finetune the model on the RIO dataset for 100 epochs with a batch size of 32. Images are resized to $512 \times 512$, and polygon augmentation is applied with a probability of 50%. The 2D coordinate embedding codebook is constructed with $64 \times 64$ bins.

## C Additional Experiment Results

### C.1 Failure Cases Study

As shown in Figure 7, we visualize some prediction failures of the TOIST model on the RIO dataset uncommon set, the blue boxes are the ground truth and the red boxes are the prediction results. When no object in the model prediction result reaches the probability threshold, the output is the empty set, *i.e.*, no prediction result, and no red box. We find that among the uncommon set prediction failure samples, (a), (c), and (h) are the failure samples where the prediction is partially correct, but not the complete prediction. Among them, (a) is a priority judgment error. In (a), both bottle and cup can be used as alternatives to the vase to mount flowers in special cases, but the bottle is more suitable than the cup and only the bottle should be predicted. (c) and (h) may be an incomplete prediction due to masking. (b), (c), (e), (f), and (g) are all samples with no correct predictions, and the model does not seem to be able to reason about these uncommon functions.

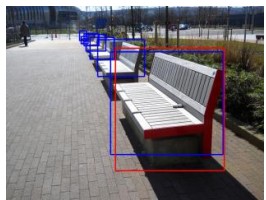 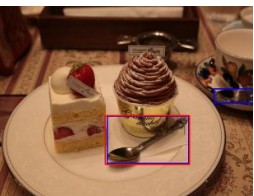 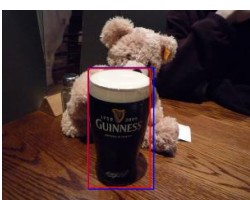 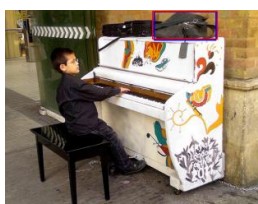

*(1) Something you can use to sleep on by placing a sleeping bag on top of it.*

*(2) Something you can use to cut the strawberries and chocolate into small pieces.*

*(3) Something you can use to fill it with water and add some fresh flowers.*

*(4) Something you can use to open it up and use it as a cover from the rain.*

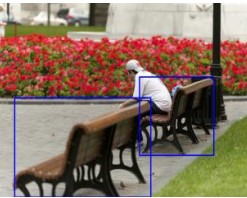 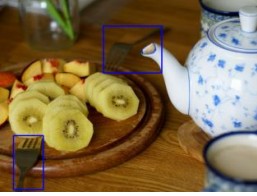 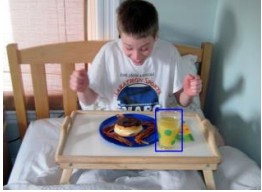 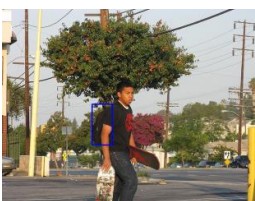

*(5) Something you can use to write, study, or for other activities.*

*(6) Something you can use to cut up the kiwi and peach pieces on the plate.*

*(7) Something you can use to fill it with flowers or other decorative items.*

*(8) Something you can use to hold over your head to keep out the rain.*

Figure 8: The case study of the uncommon set predicted by the TOIST model, the blue box is the ground truth, and the red box is the prediction result. When no object in the model prediction result reaches the probability threshold, the output is the empty set.

## C.2 Cases Study of Uncommon Set

In Fig. 8, we visualize more results in the uncommon set for observation. The blue boxes represent the ground truth, and the red boxes denote predictions.

It is discernible that for analogous tasks if the shape of the target object is similar to that of commonly observed objects, it is more readily detected. For instance, in (1), the shape of the bench is reminiscent of a bed, leading to a more accurate identification. Conversely, in (5), objects typically associated with actions like "write" and "study" are desks; given the stark contrast in appearance between benches and desks, the former is not detected.

For objects like cups, which frequently appear alongside water, detection can be influenced by context. Even though both scenarios in (3) and (7) aim to use a cup as a surrogate for a vase's function, the presence of "water" in the sentence in (3) versus its absence in (7) leads to different detection outcomes. For items with more variable appearances, such as "backpacks", the detection outcome can vary even if the intention description remains relatively consistent. For example, a horizontally placed backpack more closely resembles the shape of an umbrella, as seen in (4).

According to the detection results of the current models, it is evident that they primarily rely on appearance for judgments. Incorporating reasoning with external knowledge from large language models in the future should offer a more effective solution to challenges related to uncommon intentions.

## D Details of Data annotation

### D.1 Data Annotation Platform

We build a simple annotation platform as shown in Fig. 9. On the left is the image and the object bounding boxes, and on the right is the intention description of the object. The annotation goal is to check whether the intention description is appropriate for the object boxed out in the figure and whether there are other objects in the figure that better match the description.

The checking scores are divided into 3 grades:

**Fit-and-Best** The description is reasonable and feasible (Fit) and the object class is the best choice that will satisfy the intention (Best).

**Not-the-Best** The description is reasonable and feasible (Fit) but there are other object classes in the image that are better suited to this intention.

**Unfit** Unreasonable description or unsuitable for intention-oriented object detection.

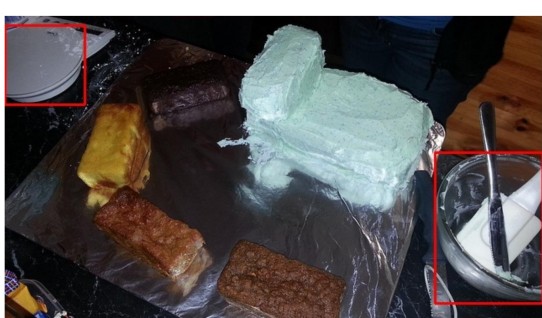

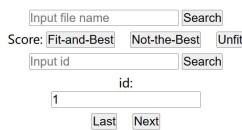

Figure 9: Screenshot of the data annotation platform.

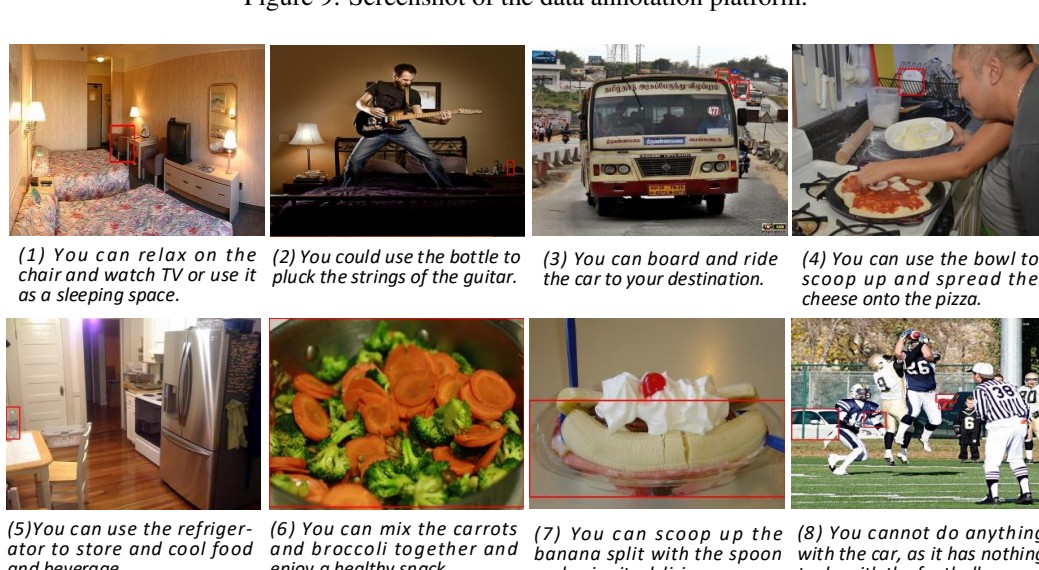

(1) You can relax on the chair and watch TV or use it as a sleeping space.

(2) You could use the bottle to pluck the strings of the guitar.

(3) You can board and ride the car to your destination.

(4) You can use the bowl to scoop up and spread the cheese onto the pizza.

(5)You can use the refrigerator to store and cool food and beverage.

(6) You can mix the carrots and broccoli together and enjoy a healthy snack.

(7) You can scoop up the banana split with the spoon and enjoy its deliciousness.

(8) You cannot do anything with the car, as it has nothing to do with the football game.

Figure 10: Typical examples of dropped data from original data.

The human annotation statistic numbers are tabulated in Table 5.

| Annotator Num. | Annotation Num. | Annotation Time | Annotation Consistency |
|---|---|---|---|
| 10 | $\sim$ 5000 per person | $\sim$10 days | 93.7% |

Table 5: Details for human annotation.

### D.2 Examples of Dropped Data

We show some typical examples of data dropped by the annotators in Fig. 10. Referring to our annotation grading, (1)-(4) are "Not-the-Best", specifically, in (1) a bed is clearly a better choice than a chair to "relax and watch TV or sleep", in (2) a bottle can be used to strum a guitar but it's not the best choice, a plectrum or finger is more appropriate, in (3) a bus would be more appropriate than a car in the back, in (4) a spoon is more appropriate than a bowl to scoop up and spread the cheese onto the pizza; (5)-(8) belong to the "Unfit" category, in (5) the boxed object is a bottle but the description is about a refrigerator, in (6) and (7), the objects are bowls, but the descriptions have nothing to do with bowls, and (8) represents an example of a part of an inappropriate description, with the sentence format like "you cannot do anything with...".

