# OpenReview forum: "RIO: A Benchmark for Reasoning Intention-Oriented Objects in Open Environments"
_NeurIPS.cc/2023/Track/Datasets_and_Benchmarks — NeurIPS 2023 Datasets and Benchmarks Poster_

### Official Review · Reviewer_gkNq · 2023-07-16
**Solid work on intention-oriented objects**

**Rating:** 7
**Confidence:** 4
**Clarity:** Yes.

**Strengths:**

The paper is well motivated and nicely written. First of all, the paper provides a strong motivation why intention-orietned object detection/segmentation is needed. The authors made clear distinction between the new task and previous similar tasks such as affordance detection and referring expression comprehension (in Fig 1 and Table 1).

The authors also did a great job of describing the step-by-step data collection process. By leveraging ChatGPT, the authors are able to curate a pretty large dataset with complete natural sentence annotation.

In terms of experimental evaluation, the metrics are pretty standard and self-explanatory. It’s appreciated that the authors build four different baselines based on previous related works to cater to their new proposed task. It would be great if the authors can also open source the training scripts / checkpoints of these baselines models.


**Additional Feedback:**

No additional feedback.


**Correctness:**

Yes, it’s constructed in a sound way and experimental evaluation is quite solid.


**Documentation:**

Yes.

**Ethics:**

No.

**Limitations:**

Included in appendix.


**Opportunities For Improvement:**


The paper has a few room for improvement. First of all, how is the proposed task different from task-driven object detection. In Fig 1, TDOD is classified as closed set, but I guess it doesn’t have to be. Task-driven object detection seems pretty similar to intention-oriented object detection as it is also scene-dependent and also have common/uncommon use case. It would be great if the authors can be more clear about the difference between the two.

Also, in experimental evaluation, clear performance drop is observed for uncommon set. Is it just because of data scarcity? It would be great if the authors can provide further insights and potential ideas to close the performance gap.

Another minor comment is that at the end of page 8, “As indicated in Table ??” is missing a reference.


**Relation To Prior Work:**

The paper has a pretty strong related work section. The authors can further differentiate intention-orientade object detection/segmentation from the task-driven one, as mentioned above.

**Summary And Contributions:**

This paper proposes a new comprehensive dataset called Reasoning Intention-Oriented Objects. The dataset offers over 100K intention-object pairs, where the intention descriptions are contextual relevant with the objects with natural sentences (rather than a mere word/verb phase) and open vocabulary. The authors also evaluate the ability of SOTA methods on the proposed reason intention-oriented objects tests in open environments.

---

> ### Author Response · Authors · 2023-08-21
>
> **Q1: Clarifying Differences Between Task-Driven and Intention-Oriented Object Detection**
> Task-driven and intention-oriented are similar to some extent. However, in some scenarios in our dataset, the term "Intention" is more appropriate than "task". For instance, "pet something" is a specific task, but when the given sentence is "pet something to seek comfort and reduce stress", it's more accurately described as an intention. Similarly, "sit on something" is a specific task, while "sit comfortably on something to relax" better represents an intention.
>
> **Q2: Is clear performance drop for uncommon set just because of data scarcity?**
> We consider imbalanced data pairing to be the main reason. In fact, there is no scarcity of data, there is sufficient data for this category of objects and similar intention descriptions, it is just that certain category intentions are not common. We visualize in the resubmitted supplementary material Sec.C.2 (Line 68-84) a comparison of success and failure examples of the same category of objects in the uncommon set with different intention descriptions, and provide further analysis.
>
> **Q3: Correcting the Missing Table Reference on Page 8**
> We have corrected the missing table reference on Page 8 in the resubmitted paper.
>
> **Q4: Open Source for the Training Scripts and Checkpoints of these Baselines Models**
> We will update in https://github.com/qumengxue/RIO before August 29th.

---

> > ### Comment · Reviewer_gkNq · 2023-08-30
> > **Thank you**
> >
> > Thank you authors for your detailed response. I will keep my original rating. Thanks again!

---

### Official Review · Reviewer_eKT9 · 2023-07-24

**Rating:** 6
**Confidence:** 3
**Correctness:** The submissison looks technically sou…
**Clarity:** Yes.

**Strengths:**

* I find this new dataset interesting to VL communities, which is potentially valuable to future research on intention-oriented tasks. It provides a large number of affordances, with intention description given in sentence.
* The authors provide sufficient details on data collection process, implementation of several baselines, and the evaluation.
* The paper is well-written and presented in a manner that is easy to follow.

**Additional Feedback:**

* In Figure 1, it seems that the text and image example for TDOD may not be properly paired together. Is this an incorrect example?

* The supplementary material should provide the specific details of prompts used for the ChatGPT APIs to ensure reproducibility. These details include the temperature, general content for the system, etc.

* The main paper / supplementary material lacks details for human annotation (the instructions to human annotators, the statistic numbers: the number of human annotators, the wage/annotation fee, etc).

* While human priority judgments are subjective, the filtering process should be carefully conducted for both the training set and evaluation sets (test & val).

**Documentation:**

* The paper is well-documented. The supplementary material provides sufficient information.
* However, there is currently no code for reproducibility.

**Limitations:**

> Have the authors adequately addressed the limitations and potential negative societal impact of their work?

Yes, they did discuss the limitations and ethical concerns in appendix A. However, I believe there should have been a more comprehensive discussion about the use of ChatGPT.

**Opportunities For Improvement:**

* As data collection is primarily automated using existing annotations from the COCO dataset, why haven't the authors considered collecting the dataset with annotations from other similar sources, such as Visual Genome?

* The quality, diversity, and human-likeness of the intention descriptions all rely on ChatGPT (with human annotators checking only). I believe the paper should include a more comprehensive discussion on these aspects, especially when the community uses it as an (human) intention-oriented benchmark.

**Relation To Prior Work:**

> Is it clearly discussed how this work differs from previous contributions?

Yes. Table 1 provides a clear comparison of the proposed dataset with other existing datasets.

**Summary And Contributions:**

* The paper introduces the concept of intention-oriented object detection and addresses the limitations in existing research by proposing a dataset called Reasoning Intention-Oriented Objects (RIO).

* RIO includes diverse real-world scenarios, a wide range of object categories, and intention descriptions represented as natural sentences. The dataset contains 40,214 images, 130,585 intention-object pairs, and more than 100 affordances. The authors evaluate existing models' ability to reason intention-oriented objects in open environments using RIO.

Overall, the paper's contribution is marginally above the acceptance threshold due to its new dataset that potentially interests the community.

---

> ### Author Response · Authors · 2023-08-21
>
> **Q1: Reasons for Collecting Data Based on the COCO Dataset**
> * COCO is more widely used in the fields of object detection and segmentation. Visual Genome (VG) doesn't have segmentation annotations and would require higher annotation costs.
> * The COCO dataset is annotated by professionals, ensuring higher quality and consistency. On the other hand, VG is crowd-sourced, which could potentially result in inconsistencies and redundancies, requiring a higher cost for data cleaning.
> * In the future, we are considering expanding our dataset, and we will look into updating our data collection based on Visual Genome.
>
> **Q2: More Comprehensive Discussion about the Use of ChatGPT**
> * Quality: ChatGPT has been trained on a vast amount of text data, enabling it to generate high-quality text outputs. We randomly sampled 500 original answers generated by ChatGPT and assessed their suitability for dataset construction based on accuracy, completeness, and fluency. The usability rate of the data was 96.2%.
> * Diversity: When using ChatGPT, the value of temperature can be adjusted to control the randomness of the output. The higher the value (closer to 1), the more varied and creative the output; the lower the value (closer to 0), the more deterministic but also potentially too repetitive the output. We set temperature=0.7 to ensure the diversity of the model output. In addition, WordCloud in Figure 4 and the dataset statistics in Figure 5 also illustrate the diversity of responses generated by ChatGPT.
> * Human-likeness: ChatGPT is already been implemented as a product and, from a language expression perspective, it can provide relatively accurate descriptions of human intentions. From the perspective of linguistic naturalness, the 500 answers randomly sampled from original answers generated by ChatGPT all meet the standards of human-likeness.
>
> **Q3: Code for Reproducibility**
> We will update in https://github.com/qumengxue/RIO before August 29th.
>
> **Q4: Clarifying a Potential Mismatch in the TDOD Example in Figure 1**
> The TDOD example in Figure 1 is matched. Both the knife and the fork can be used to "get lemon out of tea", which is task 6 among 14 tasks in the TDOD dataset. The scenes and tasks in this dataset are not related, which might give the impression of a mismatch.
>
> **Q5: The Specific Details of the Use of ChatGPT APIs**
> Our command to use the API of the ChatGPT's original model is as follows:
> ```
> response = openai.Completion.create(
>            model="text-davinci-003",
>            prompt=question,
>            temperature=0.7,
>            max_tokens=256,
>            top_p=1,
>            frequency_penalty=0,
>            presence_penalty=0)
> ```
> The question is formatted as: "Suppose you are in a place: [Caption], What can you do with the [Obj]?"
> **Q6: Details for Human Annotation**
> The annotation platform and instructions to human annotators are in Sec.D.1 (Line 86) of the Supplementary Material, and other required statistic numbers are tabulated below.
> | Annotator Num. | Annotation Num.  | Annotation Time | Annotation Consistency |
> | -------------- | ---------------- | --------------- | ---------------------- |
> | 10             | ~5000 per person | ~10 days        | 93.7%                  |
>
>
> **Q7: Training Set Filtering**
> * We appreciate the valuable feedback from the reviewer. The issue you've raised is indeed important. Our original intention was to utilize as much of the data generated by ChatGPT as possible to achieve as much diversity as possible.
> * We plan to filter our training set more carefully in future work. We will establish stricter standards and may involve human annotators for quality control. For the evaluation sets, we have already applied more careful filtering and review to ensure they can fairly assess the performance of our models.

---

> > ### Comment · Reviewer_eKT9 · 2023-08-30
> > **Official Comment by Reviewer eKT9**
> >
> > Thank you for your rebuttal which addressed several of my concerns. I suggest the authors will incorporate these detailed information by the reviewers in the main paper and supplementary material.

---

### Official Review · Reviewer_UNtV · 2023-07-25
**Review. RIO: A Benchmark for Reasoning Intention-Oriented Objects in Open Environments**

**Rating:** 6
**Confidence:** 4

**Strengths:**

• The paper addresses an interesting problem of detecting objects based on human usage.

• The paper offers a large dataset for 40,214 images and 130,585 intention-object pairs, both for training and testing the task of Intention-Oriented Object Detection. Its annotations are sentence-like and are revised by humans. This is the paper's main contribution, which required a lot of work.

• The paper tests relevant methods for the IOOD task.

**Additional Feedback:**

I am willing to increase my score if the authors can provide a clear motivation for their task and the image-text alignment as a solution to it.

**Clarity:**

Some sentences could be rephrased to sound correct or better. Please note that it's essential to maintain consistency in tense usage within a paragraph to ensure clarity and readability.

L162 “we first statistic the category functions in common instructions” - clarity

L188 “ROI task” – ROI is a dataset not a task, the tasks are IOOD and IOOS

L212 “we will also build a baseline based on it. ” -  tense

**Correctness:**

The claims seem correct. However, the experiments do not fully reveal the challenges the dataset poses. Adding the suggested baselines in the “Opportunities For Improvement” section would give the community some insights into the underlying challenges. For example, it could help determine whether the problem lies in the detection of the object or the reasoning of the affordance and object class.

**Documentation:**

There is partial documentation of the annotation process. Questions missing:

    1. How many annotators were involved in the manual selection process, and how much were they remunerated?

    2. What was the consistency and quality of annotators? What was the agreement percentage?

    3. There is no information about the hosting and maintenance plan.

**Ethics:**

No concerns

**Limitations:**

They are addressed in the supplementary material.

The usage of objects for different tasks is a subjective criterion. One person might use a backpack for cover from the rain, while a different person might not think this is useful at all (and it also depends on the backpack size). This ambiguity is certainly not discussed in the paper. How do the authors deal with classes that, depending on the instance or the subjectivity of the annotator, might change the criteria of fulfilling the task?

**Opportunities For Improvement:**

1. One question that is not completely clear is if part of the task is actually to classify the detected box into the COCO classes, or is it only supposed to detect the object that suits the usage of the caption?

    2. Figure 5 seems confusing to me. Does this mean that all verbs are associated with ~30 different categories of the COCO dataset? Are you counting instances of the same category as 2 different “category points”? Otherwise, how can verbs like "ride" have more than 40 different categories associated?

    3. In L92, one of the motivations for using sentences instead of fixed “affordances” (usage-object) is because this limits human-object interaction. Limiting the number of object categories has the same effect in this line of thinking. Annotating an infinite number of categories is infeasible, but annotating all the ones that can fulfill the “usage” within an image would be ideal and possible. Since the RIO dataset is based on COCO annotations, if a model predicts an object that fulfills the “usage” but is not within the 69 categories, it will be wrongly penalized. How do the authors propose to treat such scenarios to avoid unfair penalization?

    4. One of my main concerns in the problem formulation is whether there is actually a need for visual-text alignment. One could simply train a detector with all the categories and train independently a language reasoner to answer the questions based on the possible categories. Then, at inference time, the prediction would be simply the intersection of both sets (predicted detected objects and possible objects that fulfill the condition). This should be a reasonable baseline to include. I would like the authors to give a motivation for why combining the tasks is beneficial as compared to both tasks independently.

    5. One of the claims of the authors when including tailed (uncommon) combinations is to test the generalization of methods on rare combinations. A baseline supporting this would be simply training a language model (or using the language embedder of the proposed models) only to predict the most likely object category. This would show how much information is contained in the statistics of the dataset and dependent on the actual image instance.

    6. The paper does not provide any information on the agreement of manual annotators with ChatGPT and between the different annotators. How many descriptions were dropped after the manual selection processes? What were some common errors that were removed from the original descriptions?

**Relation To Prior Work:**

Clear

**Summary And Contributions:**

RIO proposed a dataset and benchmark for localizing (detecting and segmenting) objects, which can be used for specific purposes described by sentences. It considers images from the MSCOCO dataset and 69 object categories. Using the API of ChatGPT, they generate possible usage-describing sentences for every object. They manually filter the sentences and consider multiple object categories that could have the same usage. The task of Reasoning Intention-Oriented Objects aims to detect all objects in an image that fulfill a specific usage stated in a sentence.

---

> ### Author Response · Authors · 2023-08-21
>
> **Q1: Classify the detected box into the COCO classes or detect the object that suits the usage of the caption.**
> It's the latter. This task only requires detecting (IOOD) or segmenting (IOOS) the relevant target according to the intention description. There is no need to classify the detected object into COCO classes.
>
> **Q2: Clarification on the Interpretation of Figure 5 and Category Associations with Verbs**
> We apologize for the confusion caused by Figure 5 in the paper. Figure 5(b) does not imply that each verb is directly associated with ~30 categories. What it actually represents is the frequency of occurrence of a verb across different categories of intention descriptions. For instance, the verb 'ride' could be indirectly mentioned in the intention descriptions of other categories, such as that for an umbrella, like "something to shelter you from rain when you ride a bike".  We have recognized this ambiguity and have made clarifications in the revised version of the paper that we submitted. I hope this clears up your concerns.
>
> **Q3: Addressing Concerns About Potential Unfair Penalization in the RIO Dataset**
> In fact, not all COCO categories that can fulfill the "usage" within an image are annotated in RIO. Only the most suitable category is selected. During the data selection stage, we did not inform the annotators about the COCO categories present in the image. When an annotator is provided with an image, an intention description, and object bounding boxes, if they find there is another category in the image that better fulfills the intention description, then this data is discarded. For instance, if the description of "chair" in the image is "You can watch TV and sleep in the chair", but there is a "bed" in the image that is more suitable for "watching TV and sleeping", this data will be discarded. More examples can be found in Fig.4 of the revised supplementary material.
>
> **Q4: Baseline construction of Object Detector + Language Reasoner**
> It is feasible to train an object detector and a language reasoner independently to complete this task. We supplement the construction of this baseline and the results in the resubmitted paper, please see **Sec.4.1 (Line207)**. Considering that IOOD is in an open-set environment, which predicts the object directly based on the intention description and without predicting the class, we utilize the SOTA model for open-set object detection, GroundingDINO[1], which is also class-agnostic, as the object detector to ensure the fairness of the comparison. For the language reasoning component, we directly adopt the ChatGPT.
> [1] Liu, Shilong, Zhaoyang Zeng, Tianhe Ren, Feng Li, Hao Zhang, Jie Yang, Chunyuan Li, et al. "Grounding dino: Marrying dino with grounded pre-training for open-set object detection." arXiv preprint arXiv:2303.05499 (2023).
>
> **Q5: The Need for Visual-Text Alignment**
> * **Performance Advantages**: In terms of results, visual-text alignment is superior to the Detector+Reasoner hybrid approach. The final performance of the hybrid method can be affected by both category accuracy and reasoning accuracy. Misclassification of categories can lead to incorrect judgments when integrating with LLM.
> * **Generalization Advantages**: One perennial challenge in open-set object detection is the inconsistent labeling of the same object by different annotators. This inconsistency arises because various annotators might use different category names for the same object, making standardization a complex task. To illustrate that, we conducted a simple ablation study. By replacing original COCO category terms with commonly used synonyms, the performance of GroundingDINO on the COCO2017 validation set exhibited a noticeable decline, as shown in the table below. This observation emphasizes the motivation behind our IOOD task - our goal is to avoid intricate name definitions. Instead, the IOOD task focuses on detecting objects based on their functions and human intent, which encourages the model to understand the essence of the object, not just its categorical label.
>
> | Category Name | AP   | AP50 |
> | ------------- | ---- |:----:|
> | Original      | 48.4 | 64.6 |
> | Synonymous      | 35.6 | 47.5 |
>
> * **Structural Advantages**：Compared with end-to-end models, hybrid models are limited by the capabilities of their individual components and often struggle to utilize the combined feature information from vision and language. Furthermore, in hybrid models, there's an increased risk of error propagation. In contrast, end-to-end systems streamline these processes, optimizing the synergy between vision and language while reducing cascading errors.

---

> > ### Author Response · Authors · 2023-08-21
> >
> > **Q6: Whether the Key to the IOOD Problem is the Object Detection or the Reasoning of the Affordance and class**
> > Both are important. For object detection, in addition to the accuracy of the bounding box (IoU) affecting the final result, the wrong predicted category fed into the reasoning model also affects the reasoning result. For Affordance Reasoning, the accuracy of the reasoning result also affects the final accuracy. We randomly sampled 1000 data with ground truth categories and GroundingDINO predicted categories as candidates for ChatGPT to select according to the intention description, and the selection accuracies are shown in the table below. This indicates that the accuracy of the predicted category detected has a significant impact on Language Reasoner (11.6%). In addition, the following table also illustrates that there is still significant room for improvement in reasoning accuracy (75.5%) even when the given candidate categories are completely accurate.
> >
> > | Predicted | Ground Truth |
> > | -------- | -------- |
> > | 63.8%     | 75.4%     |
> >
> >
> > **Q7: Additional Information on Manual Annotator Agreement and Selection Process**
> > We randomly select 500 raw samples generated by ChatGPT, of which 96.2% were recognized by human annotators. Although most of the descriptions are reasonable, to ensure the uniqueness of the inference, we carefully filtered the test set, requiring that only the best choices be retained. As a result, 14,941 pairs were dropped from 44,285 pairs of raw data, and the agreement between different annotators is 93.7%.
> > We've showcased some typical examples that were removed from the original descriptions in Fig.4 (Line 97-106) of the revised supplementary material.
> >
> > **Q8: Additional Limitation Discussion of the Subjectivity of Object Usage and Its Impact on Task Fulfillment Criteria**
> > We supplemented the limitation discussion in the Limitation of the resubmitted supplemental materials.
> >
> > **Q9: Clarity Modifications**
> > Clarity modifications are included in the body of the resubmission.
> >
> > **Q10: Additional Documentation**
> > 1. 10 annotators were involved in the manual selection process. The annotators were all research members of our laboratory. They receive a standard monthly remuneration, and data annotation is a part of their regular duties. Therefore, no additional compensation was provided specifically for data annotation.
> > 2. We formulated uniform rules for annotation and performed multiple quality checks to ensure the consistency of the annotations. This included random spot checks on the annotated results and automatic checks of the annotations. We also used multiple annotators to annotate the same image for a fairer evaluation. The agreement percentage is 93.7%.
> > 3. We will update the information about the hosting and maintenance plan in https://github.com/qumengxue/RIO before August 29th.

---

> > > ### Comment · Reviewer_UNtV · 2023-08-29
> > > **Thank you for your reply**
> > >
> > > Thank you for incorporating the baseline method. It highlights that a deeper integration between detection and reasoning is necessary to tackle the task effectively. I'm curious if, based on statistics, the detector can identify objects and rely solely on the training dataset's statistics to infer the correct objects that fit the tasks.
> > >
> > > Figure 5 represents "the frequency of occurrence of a verb across different categories of intention descriptions," but it also includes instances where the verb is not related to the noun (as shown in the provided example). So, how do these plots become useful in the analysis, and what conclusions can we draw from them? I'd say that the argument in (L186), "there are an average of 158 relevant verb words," might not hold true, considering that verb-noun relationships were only counted for "relevant" instances.
> > >
> > > The issue of task ambiguity remains. Let's consider Figure 2 in the main paper. The chair in the background could also effectively fulfill the sentence "Something to lounge on and relax while reading a book." And in Figure 2 of the supplementary material, nearly any cup could be used for "Something you can use to fill it with flowers or other decorative items" or "Something you can use to fill it with water and add some fresh flowers." Why not simply identify the cups? Similarly, in Figure 2(8), for "Something you can use to hold over your head to keep out the rain," why is the backpack exclusively chosen while the two skateboards could likely serve the same purpose of rain protection?
> > > The two issues I see are: when multiple objects in the image fulfill the task, the paper 1. confines itself to coco categories, and 2. not all categories within coco that are present in the image are actually labeled as positive. Because the Reasoning Intention is a very subjective task, it makes me question your statement about "requiring that only the best choices be retained" during dataset filtering.
> > >
> > >
> > > Tense errors still persist, such as in the caption of Figure 2 in the main paper. It's written in the present and then switches to the future tense.
> > >
> > > I'm raising my score since I consider the work relevant and useful, but I'd like to suggest that the authors consider my comments to bolster the paper's strength.

---

> > > > ### Author Response · Authors · 2023-08-29
> > > >
> > > > Thank you for your response, the following is our further explanation of your question, which we hope will address your concerns.
> > > >
> > > > **Q1: Inferring correct objects based on statistics**
> > > >
> > > > Based on statistical data, it's feasible for detectors to recognize objects. While it might not be well-suited for tasks like "Intention-oriented object reasoning in open environments". To employ such a statistical approach, one would need to first consolidate the diverse open-sentence descriptions into a finite set of "intentions". Subsequently, for each defined intention, the corresponding object detection category is determined. However, this methodology presents challenges:
> > > > * Granularity: Determining the appropriate level of detail for categorizing intentions is complex.
> > > > * Classification Accuracy: As intentions can be expressed in myriad ways, ensuring accurate categorization is critical.
> > > > * New Intentions: When new requirements emerge, the capability to precisely map them to an existing category becomes pivotal, raising questions about adaptability and accuracy.
> > > >
> > > > **Q2: Analysis of Figure 5**
> > > >
> > > > From Figure 5, we can see which verbs are common and which are less so. This helps us know which verbs are mainly used and which are more specific or secondary. In addition, including instances where verbs appear unrelated to nouns is crucial because it underscores the intricacy and diversity of language. For instance, a verb might relate to multiple nouns or contexts, or it might stand independently without referencing a specific noun. This accentuates that a singular verb-noun pairing approach might not capture the complexity inherent in all intention descriptions.
> > > >
> > > > We modify the argument in (L186) as "For each category, an average of 158 verb words appear within their descriptive sentences".
> > > >
> > > >
> > > >
> > > > **Q3: Clarification on the task ambiguity**
> > > >
> > > > The first thing that needs to be clarified about our data filtering is to select the most appropriate category of objects based on a certain intentional description in a specific image scene. Therefore not all coco categories that appear are labeled as positive, only the most appropriate ones are labeled. Second, since our data is constructed based on the coco dataset, the selected objects are limited by the category of coco. In the future, we will expand the RIO data on datasets with more categories, such as the Lvis dataset.
> > > >
> > > > For the several examples you mentioned, the corresponding explanations are as follows:
> > > > * In Figure 2 of the main paper, the sofa is more comfortable than the chair for "lounge on and relax", so choose "sofa" rather than "chair".
> > > > * In Figure 2 of the supplementary material, we just show whether the alternative object "cup" can be detected given the intention of "fill it with flowers or other decorative items" without the common object "vase". When there is neither a vase nor a cup in the scene, it is also possible to detect the presence of bottles or other containers. Therefore, it is not suitable to recognize a "cup" simply by seeing the intention "fill it with flowers or other decorative items". The key to solving the intention-oriented task is to locate the object that best matches the intention description in a given scene.
> > > > * In Figure 2(8), it is indeed a matter of preference, as at least two of the annotators thought that a "backpack" would be more suitable than a "skateboard" to "hold over your head to keep out the rain". Skateboards are likely to be heavier, and when it rains, skateboards may be more appropriate for accelerating to a destination while backpacks may be more appropriate for keeping the rain out. People do have a variety of choices, and we have tried to select the more popular categories for evaluation as much as possible.
> > > >
> > > > **Q4: Tense Errors**
> > > >
> > > > Thank you for identifying the tense errors, we will carefully correct the tense errors in the paper.

---

### Decision · Program_Chairs · 2023-09-22

**Decision:**

Accept (Poster)

**Comment:**

All reviewers agree on the acceptance of the paper. Also, reviewers acknowledge the rebuttal and consider most of their concerns alleviated. After reading the reviews, rebuttal, and the discussion, I think that the paper is relevant and useful to the community, and propose to accept it. I do think authors should add the comments from the reviewers in the final version, and especially correct the grammar errors.